# Prevalence of Burnout among Teachers during the COVID-19 Pandemic: A Meta-Analysis

**DOI:** 10.3390/ijerph20064866

**Published:** 2023-03-10

**Authors:** Naiara Ozamiz-Etxebarria, Idoia Legorburu Fernnadez, Darren M. Lipnicki, Nahia Idoiaga Mondragon, Javier Santabárbara

**Affiliations:** 1Department of Developmental and Educational Psychology, University of the Basque Country UPV/EHU, 48940 Leioa, Spain; 2Centre for Healthy Brain Ageing, University of New South Wales, Sydney, NSW 2052, Australia; 3Department of Microbiology, Pediatrics, Radiology, and Public Health, University of Zaragoza, C/Domingo Miral s/n, 50009 Zaragoza, Spain

**Keywords:** burnout, teachers, COVID-19, prevalence, meta-analysis

## Abstract

Since the start of the COVID-19 pandemic, many studies have found that there has been a lot of teacher overload. One of the additional burdens has been that they have had to teach online. In addition, when they returned to face-to-face classes, they had to follow all the hygiene rules so that the COVID-19 virus would not spread. It is therefore not surprising that, during this pandemic period, high levels of psychological symptoms have been reported among teachers. Among this symptomatology, burnout has been very frequent among teachers. Therefore, the aim of this study is to conduct a meta-analysis to determine the overall prevalence of burnout among teachers during the COVID-19 pandemic. For this purpose, a search was conducted for cross-sectional studies listed in PubMed from 1 December 2019 to 14 February 2022 that reported on the prevalence of burnout among teachers. A total of nine studies from eight different countries in Africa, Asia, Europe, and North and South America, were included in this study. The pooled prevalence of burnout among teachers was 52% (95% CI 33–71%), which is higher than burnout rates reported for health professionals. There was significant heterogeneity between studies (*I*^2^ = 99%, *p*-value < 0.001), and the prevalence of burnout was higher in women and school teachers (compared to university educators), and lower in American studies. This meta-analysis concludes that teachers worldwide experienced a high rate of burnout during the COVID-19 pandemic. This has implications not only for the teachers themselves, but also for the quality of the education they were able to provide. This education has an influence on the student population. The possible long-term effects are yet to be determined.

## 1. Introduction

It has now been 3 years since the World Health Organization declared a pandemic in March 2020 due to the spread of the COVID-19 virus [1,2]. Since the beginning of the pandemic, many measures have been taken to stop the spread of the virus. Some of the measures included lockdowns. Many people had to be locked in their homes for long periods of time. Other restrictions were strict distancing measures under which people could not get close to people, to stop the spread of the virus [3,4,5]. This situation has brought a lot of fear and uncertainty, creating psychological symptomatology in all countries of the world and in all types of populations [6,7]. The psychological symptoms have been varied, such as stress, anxiety, depression, or burnout. This is why many studies have been carried out all over the world to study the psychological consequences of the pandemic on the general population, as well as on the most affected groups, such as health and education professionals, among others. There is still great concern about the aftermath of the pandemic in post-pandemic times.

The closure of schools and universities was one of the first measures taken by governments to stop the spread of the virus in many countries [8,9]. The closure of these centers led to the delivery of classes in an online mode [10] at the beginning of the pandemic, followed by a bimodal mode [11,12] where some of the class was face-to-face while others followed the classes from home. When educational centers finally reopened, face-to-face teaching had new challenges, including strong hygienic measures and harsh social distancing protocols [13]. The adaptation to online teaching followed by face-to-face teaching with COVID-safe protocols was often undertaken without adequate preparation or skills [14]. This has led to psychological stress, worse levels of well-being, burnout, and negative emotions [15], as well as a high prevalence of anxiety, depression, stress, and even post-traumatic stress disorder among teachers. Hygiene measures in many countries consisted of having to clean hands before entering and leaving any public place and the use of masks, as well as continuous cleaning of public places. Ventilation of enclosed places at short intervals was also important. As for harsh social distancing protocols, people had to keep meters away from each other, and many had to be locked up for different periods of time because of suspected COVID-19 or COVID-19 symptoms or when the situation in their country was one of many cases of COVID-19. Given the variety and extent of the psychological symptoms experienced, including burnout, in the general population and by teachers as a result of the COVID-19 pandemic, and those they were already suffering from, a higher than usual rate of burnout was to be expected in all jobs and also in the teacher’s job. Burnout is usually defined as a prolonged response to chronic emotional and interpersonal stressors, characterized by emotional exhaustion, depersonalization, and lack of social fulfilment [16]. It is a state of psychophysical exhaustion, deterioration of relationships, and a sense of professional ineffectiveness and disillusionment that can occur in a wide range of work contexts. Employees become cynical about their work and experience a decrease in professional efficacy [17].

Although the pandemic may have aggravated this symptomatology, burnout was already a global health problem among teachers before the pandemic for various reasons or multi-causal explanations [18]. Among the reasons that may provoke burnout in teachers, it is remarkable that they are employees who are responsible for the academic performance of students and therefore they have a high attached mental charge. In fact, teaching is a profession that in recent times has been degraded in many aspects, such as loss of status [19]. Moreover, political and social pressures to achieve higher quality education outcomes have increased stress and burnout among teachers [20]. An imbalance between the demand for better outcomes and inadequate resources has contributed to burnout (Baka, 2015), as have inappropriate and challenging student behaviors [21] and an overwhelming amount of paperwork [8]. Moreover, in the case of COVID-19, teachers had to suddenly pivot to digital education. This entailed the forced learning of a new way of teaching and was, across the board, a source of stress for a large portion of teachers [22]. This situation of education professionals has consequences for students. Some previous studies suggest that students with burned-out teachers tend to have less motivation for schoolwork [23] In addition, other research indicates that the psychological state of teachers affects the academic results of children and adolescents, obtaining worse academic results and more difficulty in school success [24].

Burnout is one of the major psychological problems among teachers [25]. Many teachers were already suffering from burnout before the pandemic and this situation is likely to have worsened after the pandemic for multiple reasons. It is therefore important to investigate the level of burnout among teachers during the pandemic. However, to the best of our knowledge, no meta-analysis has been conducted on the prevalence of burnout among teachers during the COVID-19 pandemic. The present study aims to find out how much burnout teachers have experienced during the pandemic in different countries.

## 2. Materials and Methods 

This study was conducted in accordance with the PRISMA guidelines for reporting systematic reviews and meta-analyses [26] (Appendix A).

### 2.1. Search Strategy

In accordance with the Campbell Collaboration [27], two researchers (JS and IL) searched for all cross-sectional studies reporting the prevalence of burnout among teachers published from 1 December 2019 through 14 February 2022, using MEDLINE via PubMed. The search terms were: (“School Teachers”[Mesh] OR “Faculty”[Mesh] OR teacher*[tiab] OR professor*[tiab] OR lecturer*[tiab] OR instructor*[tiab]) AND (Burnout, Psychological[Mesh] OR burnout[tiab])

No language restriction was made. References from selected articles were inspected to detect additional potential studies. We also manually searched the “grey literature” (e.g., medRxiv and Google Scholar) to detect other potentially eligible investigations [28]. Any disagreement was resolved by consensus among third and fourth researchers (NO-E and NI), according to Harrer et al. [28].

### 2.2. Selection Criteria

Studies were included if they: (1) reported cross-sectional data on the prevalence of burnout, or sufficient information to compute this, conducted during the COVID-19 outbreak; (2) focused on teachers; (3) included a validated instrument to assess burnout; (4) had the full text available.

We excluded studies focusing only on community-based samples of the general population or specific samples that were not teachers (e.g., students, medical professionals, patients), as well as review articles.

A pre-designed data extraction form was used to extract the following information: country, sample size, proportion of women, average age, response rate and sampling methods, the instruments used to assess burnout, and prevalence rates.

### 2.3. Methodological Quality Assessment

We assessed the quality of studies using a risk of bias tool proposed by Loney et al. [29]. Quality assessments were based on 8 criteria, each scored 0 or 1: (1) random sample or the complete population was used; (2) there was an unbiased sampling frame (i.e., census data); (3) there was an adequate sample size (>300 subjects); (4) standard measurements were used; (5) the outcomes were measured by unbiased raters; (6) there was an adequate response rate (>70%) and a description of losses; (7) confidence intervals and subgroup analysis were reported; and (8) the study subjects were described. The total score ranged from 0 (poor quality) to 8 (high quality), and studies were classified as low (6–8), moderate (4–5), or high (0–3) risk of bias.

Any disagreements between the reviewers were resolved through discussions, or by further discussion with third and fourth researchers (NO-E and NI) [28].

### 2.4. Data Extraction and Statistical Analysis

A generic inverse variance method with a random effects model was used [30], with double arcsine transformation of proportion to account for the variability and heterogeneity of prevalence rates among the included studies [31]. The main outcomes are presented in proportion format with a corresponding 95% confidence interval (95% CI) along with statistical heterogeneity results. The Hedges *Q* statistic was used to assess heterogeneity across studies, with statistical significance set at *p* < 0.10. The *I^2^* statistic and 95% CI were also used to quantify heterogeneity [32]. Values between 25–50% are considered as low, 50–75% as moderate, and 75% or more as high [33]. Heterogeneity of effects between studies occurs when differences in results for the same exposure–disease association cannot be fully explained by sampling variation. Sources of heterogeneity can include differences in study design or in demographic characteristics. We performed subgroup analyses to explore the sources of heterogeneity expected in meta-analyses of observational studies [33]. Meta-regression was not performed, as having less than 10 studies conveyed a lack of statistical power [34]. We conducted a sensitivity analysis to determine the influence of each individual study on the overall result by omitting studies one by one.

Conventional funnel plots to assess biases in meta-analyses are inaccurate for proportion studies [35], with the fail-safe N value approach better representing publication bias [36]. This statistic is recommended for meta-analyses with fewer than 10 studies [37,38], and indicates the number of non-significant, unpublished (or missing) studies that would need to be added to the meta-analysis to reduce an overall statistically significant result to non-significance. There is confidence in the summary conclusions if this number is large relative to the number of observed studies [36].

All statistical analyses were conducted by one researcher (JS) using R [39] with the *metaprop*, *metafor*, and *dmetar* packages for meta-analysis; *p*-values are reported as two-sided, with 0.05 accepted as statistically significant except where otherwise indicated.

## 3. Results

Figure 1 is a search strategy and study selection process flowchart. A total of 504 records were initially identified from Medline via PubMed, with 413 of these excluded after screening the titles and abstracts. Two extra records were added after a manual search of another source (Google Scholar). After reading the remaining 93 articles in full, we finally included nine in our meta-analysis [40,41,42,43,44,45,46,47,48]. Exclusion reasons are detailed in Figure 1.

Table 1 provides an overview of the general characteristics of the nine studies included in the meta-analysis, as well as their methods of measuring the outcomes and reported prevalence of burnout. Five studies used the Maslach Burnout Inventory (MBI), two used the Copenhagen Burnout Inventory (CBI), and one each used the Burnout Assessment Tool (BAT) and Professional Fulfillment Index (PFI). The teacher samples were from eight different countries and all worked across all levels of education, from primary school to university. The sample size ranged from 51 to 1372 participants, and the mean age ranged from 38.6 to 48 years. All studies included both men and women, and the percentage of women ranged from 37.5% to 98.2%. 

Our quality scores for the studies varied from 3 to 5 out of a possible total of 8 (Table 2). Only one study has a random sample [44]. The main limitation of all studies was the outcomes not being measured by unbiased raters. In addition, only one study reported the response rate, which was 17% [45]. 

Nine studies reported prevalence of burnout data, which ranged from 22.1% to 85% (Table 1). Our estimated overall prevalence of burnout was 52% (95% CI: 33–71%), with significant heterogeneity between studies (*Q* test: *p*-value < 0.001; *I*^2^ = 99%) (Figure 2). 

Our meta-regression showed that the prevalence of burnout was independent of mean age at baseline (*p* = 0.367) or methodological quality (*p* = 0.752). However, studies with a larger percentage of women reported a higher prevalence (b = 0.009; *p* = 0.013).

Our subgroup analyses to identify sources of heterogeneity found a higher prevalence of burnout for studies in Asia or Africa (71% [95% CI: 54–86%]) and Europe (68% [95% CI: 57–79%]) compared to those in America (32% [95% CI: 23–42%]). We also observed a higher prevalence of burnout for studies using the CBI (64% [95% CI: 25–94%]) or MBI (57% [95% CI: 33–80%]) compared to those using either the PFI or BAT (29% [95% CI: 4–64%]) and for studies focused on school teachers (70% [95% CI: 37%–95%]) compared with studies focused on university teachers (47% [95% CI: 22%–73%]); however, this difference did not reach statistical significance. There were insufficient data to perform subgroup analyses of sampling method or response rate.

Excluding studies one by one from the analysis did not substantially change the pooled prevalence of burnout, which varied between 48% (95% CI: 32–64%), with Xu et al. [47] excluded, and 56% (95% CI: 37–74%), with Pereira et al. [45] excluded (Figure 3). This indicates that no single study had a disproportional impact on the overall burnout prevalence.

An absence of publication bias was indicated by a fail-safe N of 21,587, indicating that 21,587 studies with null results would be needed to reduce the observed overall prevalence to non-significance.

## 4. Discussion

### 4.1. Summary of Main Findings

The COVID-19 pandemic is having an unprecedented impact on teachers, and the present study provides an up-to-date meta-analysis of nine studies reporting the prevalence of burnout among teachers during this time. Teachers, like the rest of the population, have had to face a very complicated situation that can have psychological consequences. In addition, they have had to go through different phases of adaptation to this new and difficult situation. First, they had to teach online, then they had to adapt to bimodal training, and finally, they had to adjust their classes to the hygienic measures strictly imposed to stop the spread of the COVID-19 virus. 

To the best of our knowledge, this is the first review to report overall prevalence rates of burnout among teachers across different ages, genders, countries, and educational sectors. We found that teachers report high levels of burnout, with an overall prevalence of 52%, though with significant heterogeneity among studies. These levels of burnout are higher than those found by Garcia-Arroyo et al., 2019 [18] among teachers in a meta-analysis before the pandemic. This is probably because the pandemic situation has worsened the burnout symptomatology among teachers due to the challenges they have had to face [49,50,51]. With respect to other studies conducted during the pandemic, the present results are higher than rates for healthcare professionals overall of 21% [52], and for nurses of 34.1% [53], groups that have been studied more extensively [54]. This is remarkable, as several studies have been concerned with the mental state of healthcare workers [55,56,57,58], but the mental state of teachers has not been investigated as much. Finally, the levels of burnout among teachers are also higher than the levels of anxiety (17%), depression (19%), and stress (30%) found in a previous meta-analysis of teachers during the pandemic. This makes us realize that perhaps the focus should be on burnout in teachers rather than on other more clinical symptoms. Burnout among teachers was already a problem before the pandemic [19,59,60] and it seems that the pandemic may have worsened it.

We found that the prevalence of burnout is independent of age. Previously reported findings have been mixed, with some studies reporting greater prevalence of burnout among younger, less experienced teachers [61,62], and others reporting more burnout among older teachers due to an accumulation of fatigue [63,64] or personal conflicts [65] over the years. During the pandemic, therefore, among teachers, age has not affected burnout, nor have other psychological symptoms such as stress, anxiety, or depression, as also indicated by the research of [66]. This differs from studies carried out with more general populations, where more psychological symptomatology has been found among younger people than among older people [67,68]. This may be because the pandemic has affected the whole population and it may be that the problems faced by the young and the old are the same and therefore they have experienced the same burnout symptomatology. Young people have had to face the challenges they already had as beginners added to those of the pandemic. For the older ones, however, some of them already had a previous burnout and the pandemic has added to this symptomatology [69,70,71,72,73]. Therefore, neither the older nor the younger ones suffered more, but all of them equally, as the present study shows.

Previous studies have reported that burnout is similar among men and women teachers [18,74], but we found that studies with a larger percentage of women reported a higher prevalence of burnout. This is in line with pre-pandemic studies reporting women teachers as more likely to suffer burnout [75,76,77] and with studies conducted on mental health during the pandemic in which women across all sectors showed more psychological symptoms than men [78]. It has been claimed that the pandemic has widened the gender gap, with women burdened with higher levels of care (for children, elderly, etc.) than men [79], and that women are more likely to suffer burnout in non-egalitarian societies [18,80]. Other more general studies measuring stress during pandemics in society also point to gender differences, with women being more affected [81]. The main reason may be the tendency in many cultures for women to be more involved in caring for dependents. Therefore, in addition to the work overload, women have had to cope with the work they have had to carry out in their families, which is why they may have suffered more burnout than men [79,82,83].

Our finding of a higher prevalence of burnout among school teachers than among university teachers is consistent with research from both before [84] and during [85] the pandemic. Teachers at university probably do not have the same level of COVID-safe duties as teachers at pre-, primary, or secondary schools. This extra burden might contribute to the higher rates of burnout among school teachers, as might their typically greater levels of direct interaction with both students and parents [86,87]. 

In this respect, some of the typical factors affecting burnout in school teachers continue or increase in the COVID-19 era. Workload or pedagogical barriers [60] are two typical teacher stress factors that may have caused greater difficulties for school teachers than for university teachers during the pandemic. In contrast, one of the factors that generates stress in university teaching staff is contact with students [59], whereby the distance established by the pandemic itself reduces the presence of this stress factor during this period.

It is difficult to disentangle the reasons for the differences in burnout prevalence we found between geographical regions. For example, the study we included from China reported the highest prevalence of burnout among all studies, but also had the greatest proportion of women teachers (98%). As discussed above, women generally showed a higher prevalence of burnout than men. China was also the first country affected by the pandemic. There could also be effects associated with differences between countries or regions in the educational system, including job quality and organizational characteristics [88,89,90]. Other reasons that could affect the different burnout levels between countries are the differences in the culture that involves education. The levels of participation in decision-making, role conflict, freedom and autonomy, or social support networks could be decisive factors for higher stress levels [91]. Therefore, it could be that in China, in addition to the pandemic, there were already pre-pandemic causes for more burnout among teachers.

### 4.2. Strengths and Limitations 

We believe our study to be the first meta-analysis of burnout among teachers. A significant strength is the geographical range of included studies, covering countries from the East and West, as well as both developed and developing nations. This gives confidence in our findings being globally representative. 

A limitation of our study is the variety of burnout scales used by the studies. Studies using the PFI or BAT had a lower prevalence of burnout than studies using the CBI or MBI, but the extent to which this is related to the scales themselves or factors associated with the sample and country in which they were used is unclear. It is also the case that all of these scales are self-report, and different results might be found if assessment of burnout was based on clinical interview. Further, having used cross-sectional data, we are unable to show if burnout rates differed at different stages of the pandemic. Longitudinal studies are needed to understand this, as well as any long-term effects of the pandemic on burnout among teachers. 

## 5. Conclusions

The pandemic has taught us different lessons, such as that countries with weaker financial situations have had more difficulties in implementing distance learning. In addition, teachers have had to face new challenges in online teaching [92]. It has also taught us that education systems around the world must be renewed to ensure innovative and inclusive quality education [93]. However, while it has brought lessons that we can learn from, it has also brought psychological suffering such as burnout.

This meta-analysis shows that a high proportion of teachers suffered from burnout during the COVID-19 pandemic. Indeed, the reported rate is even higher than that for healthcare professionals, including nurses, and any effects on the future of the teaching profession remain to be determined. Interventions to assist teachers with the effects of burnout during the pandemic may still be needed, possibly with tailoring to specific countries on the basis of factors such as the proportion of teachers who are women, as women seemed especially vulnerable to burnout during the pandemic. This has obvious implications not only for teachers themselves, but also for the quality of education they are able to provide to their students. 

Preventive and informational work is essential as a method of prevention, as are techniques that have proven effective in the field of psychological intervention for those cases where a disorder has already developed. This is why we believe that it is important that good practices be implemented in different schools in different countries in order to improve the mental health of teachers. 

Currently, most schools lack these resources, or the resources they have are oversaturated. However, in view of this meta-analysis, it is important that these good practices be implemented in all schools.

After all, the mental state of the teachers can influence the mental state of the students and that is why it is important to take care of it.

We propose that there should be a screening protocol for teachers on an annual period, so that they can be aware of their problems/weaknesses and address them at an early stage. It would also be important to have a psychological service to help them.

In addition to more resources in the Psychological Services, there should also be a more pleasant environment in all senses of the word. It would be interesting to look for a combination of individual therapy and group formats. It would also encourage the creation of self-help groups, fostering relationships.

Considering that psychological symptoms usually tend to worsen without help, it would be important to prevent this burnout situation among teachers from getting worse and to put resources in place as soon as possible so that they can take care of their mental health.

To deal more specifically with burnout, human resource managers in the field of healthcare must be aware that the first measure to avoid burnout syndrome is to train staff to know their symptoms. However, in addition to considering programs that involve the acquisition of knowledge, intervention attempts must incorporate other actions. Intervention strategies should consider three levels:(a)Consider the cognitive processes of self-assessment of professionals, and the development of cognitive behavioral strategies that allow them to eliminate or mitigate the source of stress, avoid the experience of stress, or neutralize the negative consequences of that experience (individual level).(b)Promote the formation of social skills and social support of professional teams (group level).(c)Eliminate or reduce the stressors of the organizational environment that give rise to the development of the syndrome (Organizational level).

At the individual level, the use of controlling or problem-focused coping strategies prevents the development of burnout syndrome. At the group and interpersonal level, the strategies include promoting social support from peers and supervisors. This type of social support should offer emotional support, but also include periodic evaluation of professionals and feedback on their development in the role. Finally, at the organizational level, the management of organizations must develop prevention programs aimed at improving the environment and climate of the organization.

## Figures and Tables

**Figure 1 ijerph-20-04866-f001:**
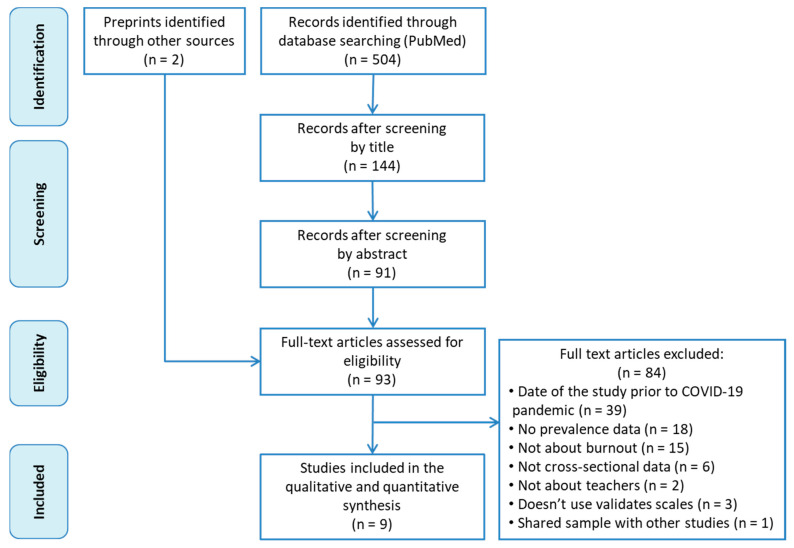
Flowchart of the study search and selection process.

**Figure 2 ijerph-20-04866-f002:**
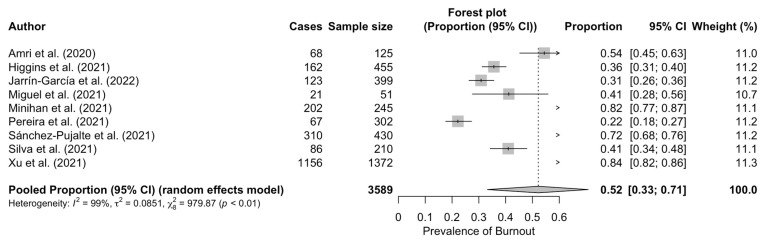
Forest plot for the prevalence of burnout among teachers [40,41,42,43,44,45,46,47,48].

**Figure 3 ijerph-20-04866-f003:**
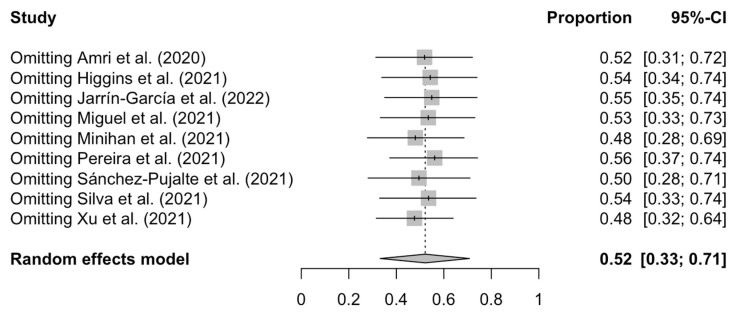
Sensitivity forest plot for the prevalence of burnout [40,41,42,43,44,45,46,47,48].

**Table 1 ijerph-20-04866-t001:** Characteristics of the studies included in the meta-analysis.

First Author (Publication Year)	Sample Country	Population	Sample Size (n)	Mean Age (SD)	% Females (n)	Response Rate (%)	Sampling Method	Burnout Scale	No. Cases (Prevalence, %)	Quality Assessment
Amri et al. (2020) [40]	Morocco	School teachers	125	38.6 (9.9)	56.8% (71)	NR	NR	MBI	68 (54%)	3
Higgins et al. (2021) [41]	United States	University teachers	456	NR	37.5% (171)	NR	Convenience	PFI	162 (37.4%)	4
Jarrín-García et al. (2022) [42]	Ecuador	University teachers	399	NR	47.9% (191)	NR	Convenience	MBI	123 (30.83%)	5
Miguel et al. (2021) [43]	Portugal	University teachers	51	48 (1)	68.6% (35)	NR	Convenience	CBI	21 (41.2%)	3
Minihan et al. (2021) [44]	Ireland	School teachers	245	44 (10.23)	89% (224)	NR	Cluster	CBI	202 (82.4%)	5
Pereira et al. (2021) [45]	Brazil	All teachers	302	46.75 (11.02)	55% (166)	17%	Convenience	BAT	67 (22.1%)	4
Sánchez-Pujalte et al. (2021) [46]	Spain	School teachers	430	41.40 (11.07)	46.28% (199)	NR	Convenience	MBI	310 (72.1%)	4
Silva et al. (2020) [47]	Brazil	University teachers	210	NR	54.8% (115)	NR	NR	MBI	86 (40.9%)	4
Xu et al. (2021) [48]	China	University teachers	1372	NR	98.2% (1348)	NR	Convenience	MBI	1156 (85%)	4

Abbreviations: SD, standard deviation; NR, not reported; MBI, Maslach Burnout Inventory; PFI, Professional Fulfillment Index; CBI, Copenhagen Burnout Inventory; BAT, Burnout Assessment Tool.

**Table 2 ijerph-20-04866-t002:** Quality assessment.

Study	1	2	3	4	5	6	7	8	Total
Amri et al. (2020) [40]	0	0	0	1	0	0	1	1	3
Higgins et al. (2021) [41]	0	0	1	1	0	0	1	1	4
Jarrín-García et al. (2022) [42]	0	1	1	1	0	0	1	1	5
Miguel et al. (2021) [43]	0	0	0	1	0	0	1	1	3
Minihan et al. (2021) [44]	1	1	0	1	0	0	1	1	5
Pereira et al. (2021) [45]	0	0	1	1	0	0	1	1	4
Sánchez-Pujalte et al. (2021) [46]	0	0	1	1	0	0	1	1	4
Silva et al. (2020) [47]	0	1	0	1	0	0	1	1	4
Xu et al. (2021) [48]	0	0	1	1	0	0	1	1	4

Abbreviations: (1) Random sample or entire population; (2) Unbiased sampling frame (census data); (3) Adequate sample size (>300 subjects); (4) Standard measures were used; (5) Outcome measured by unbiased raters; (6) Adequate response rate (>70%) and description of losses; (7) Confidence intervals and subgroup analysis; (8) Study subjects described.

## Data Availability

The data can be obtained by contacting the authors.

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
