# Peer review of "Prevalence of Burnout among Teachers during the COVID-19 Pandemic: A Meta-Analysis"

_ijerph, 2023, doi:10.3390/ijerph20064866_

Round 1
Reviewer 1 Report
This sound paper may be the first review regarding prevalence of burnout in school, secondary and university teachers across different countries. The methodology is well explained (and applied) and the results are correctly presented.
However, I have some suggestions for the authors in order to improve – in my opinion – the quality of the paper.
Firstly, in lines 39-41, the explanation of the psychological symptomatology is strictly causal (this situation has brought...., creating....), as well as in line 58 (as a result of...) when, in page 2, it is said that Burnout was already a global health problem among teachers before the pandemic (line 65).
In my opinion, a multi-causal explanation would be appropriate (situation before COVID pandemic, lack of interpersonal interaction with colleagues, lack of digital resources in some countries, etc.). In fact, in line 72, the suddenly change to digital teaching is mentioned as one of the causes of the psychological symptomatology.
Perhaps, in this way, it could be easier analyzing the reasons of rate differences in burnout prevalence between geographical regions (lines 258-259).
I appreciate as valuable the suggestions which appear in the conclusive section to both prevent and reduce the burnout situation, specially the creation of groups fostering relationships.
Author Response
Dear Reviewer
Thank you very much for reviewing the article.
Thanks to your revisions we believe that the article has been substantially improved.
We have responded to each of your revisions below and have made the changes to the article with "change control".
Thank you very much for your feedback.
As you have pointed out in the introduction, we have underlined that the reasons for burnout symptoms can be multi-causal, both pre-pandemic and post-pandemic.
We have also analysed this reason for the prevalence of burnout in different countries in the discussion as you indicate.
Reviewer 2 Report
Dear authors,
The Introduction to the article needs to be improved. I would like to see more definition of terms and very specific descriptions, for example, what is meant by: strong hygienic measures, psychological symptomatology.
After reading the introduction, it is not clear that teacher burnout is related to the other psychological symptomatology mentioned in the introduction. This link needs to be described in more detail. In my opinion, the Introduction should be strengthened by linking it to the impact of burnout on teachers as employees and on students' academic performance. At present, the Introduction is more concerned with the historical context of the pandemic than with the scientific issue of burnout. After reading the Introduction, it remains unclear why this meta-analysis was necessary? What is the authors' research objective?
I also think that the psychological consequences are being taken too broadly, reaching even into clinical mental disorders. There should be more definition of what the article is dealing with and/or whether it is within the competence of the authors to mention clinical mental disorders.
The Discussion needs to be broadened, as it is currently too narrow. It mentions the findings of meta-analyses but does not go deeper into why this is so. For example, why burnout is more common in women, why burnout differs between different age groups (what does the scientific literature say about this?).
More detailed practical recommendations would enrich the article.
The formatting of the text needs to be improved, as there are missing spaces, etc. Formatting errors have also been noted (to mention just a few), e.g. in line 74, or the absence of spaces in lines 76 and 78, and the different fonts of the tables, please refer to Figure 2.
Regards
Author Response
Dear Reviewer
Thank you very much for reviewing the article.
Thanks to your revisions we believe that the article has been substantially improved.
We have responded to each of your revisions below and have made the changes to the article with "change control".
Thank you very much for your feedback.
- We have better described the terms you indicate in the article.
- We have improved the introduction with the advice you have given us.
- In the introduction, we made it clear that burnout is a symptomatology that existed before the pandemic and also during the pandemic.
- We have also reinforced the symptomatology of burnout by relating it to the impact of burnout on teachers as employees and on students' academic performance as you have recommended.
- We have also better defined what the article is about.
- We have further deepened the discussion by providing more explanations and references.
- We have also deepened in the conclusion section.
- We have also improved the formatting of the text
Reviewer 3 Report
Thank you very much for the opportunity to review this paper. This study explores the prevalence of burnout among teachers during the COVID-19 pandemic. This is a very interesting a current topic, which could be hosted by this journal.
At the same time, there are some important issues, which require the authors' attention:
a) Abstract - I missed an explicit presentation of the purpose of this research. The words "methods" and "results" are not needed. Please keep it without titles.
b) Introduction. The authors need to reflect on the lessons learned from the pandemic and the challenges for teachers. I propose the integration of a couple of current (2022 and 2023) and credible resources such as:
https://scholarworks.waldenu.edu/hlrc/vol12/iss0/7/
https://www.tandfonline.com/doi/full/10.1080/0305764X.2022.2159013?casa_token=Wjom_kTEUDsAAAAA%3AnmaY-MkwWCuf4smg7cYIbeoZXHQz0KZ2ILTvajegTqoLkfBDyJqAoZ6gnMWGrHJAQl2It_s3mOFw
These resources have additional literature which may be useful. I believe it is not needed anymore offering introductory information about the pandemic and the lockdowns. Now it is time to reflect on this, so please provide the readers with reflection on what happened and how we can capitalize the lessons learned.
In the introduction I expected some reflection on the burnout in the workplace in general, so we can have a smooth transition to the teaching profession.
c) Academic integrity. After checking this article in Turnitin identified very high similarity percentage with other publications. Please make sure you properly paraphrase and redevelop the text, especially since you are submitting in the same journal.
d) Conclusions. The conclusions need to be further expanded. I see many very short sentences, which need further development and reflection on the implications.
Author Response
Dear Reviewer
Thank you very much for reviewing the article.
Thanks to your revisions we believe that the article has been substantially improved.
We have responded to each of your revisions below and have made the changes to the article with "change control".
Thank you very much for your feedback.
- In the abstract we have removed the subtitles as you have indicated and we have explicitly written the objective of this study.
- Thank you for the articles you have provided us with. We have added information and references to the two articles in the manuscript.
- We have also included in the introduction that burnout has occurred not only among teachers but also in general in all jobs.
- Academic integrity: thank you very much for this comment. We made changes to the paragraphs to make it totally different from other articles. We told the editors that some aspects of methodology were difficult to change since the meta-analysis expert, Dr. Santabarbara tends to use similar terms in all articles because that is how the methodology is written.
The journal made us change paragraphs and accepted the current article as written. Thank you
-We have also deepened in the conclusion section.
Reviewer 4 Report
Paper is a good study of the burnout caused by the Co-vid epidemic. The list of references is pretty exhaustive since the authors only wanted to go back a few years. The analysis is very good and the research methodology is done very well.
The grammar is very good and I only had a couple of questions:
On page #2, line #71, there a double space and then the authors are listed instead of the reference citation number. Baka 2015 is not included in the Reference list.
On the Reference list, #40, 41, 42, 43, 46, 48, and 49 are not cited in the paper. The authors are used in the list of the studies included.
The chare is very helpful.
Author Response
Dear Reviewer
Thank you very much for reviewing the article.
Thanks to your revisions we believe that the article has been substantially improved.
We have responded to each of your revisions below and have made the changes to the article with "change control".
Thank you very much for your feedback.
Thank you very much, we have changed the typos you have told us about and we have also detected others and changed them.
We have also rechecked all the bibliography to make sure it is correct.
Thank you very much
Round 2
Reviewer 2 Report
The authors have taken note of the comments. However, the presentation of the text needs to be corrected, for example in line 18 we see such signs together .:
In line 99 there is both text and title.
In some places there are no spaces in the text, e.g. 237, 241 and 246... I won't go on.
Author Response
Thank you very much for the review.
We have changed what you say and other errors that we have detected.
Kind regards
Reviewer 3 Report
The authors addressed the issues I raised in my previous feedback and I would like to thank them for considering all my recommendations.
Author Response
Thank you